# Some performance considerations when using multi-armed bandit algorithms in the presence of missing data

Xijin Chen[1]*, Kim May Lee[2], Sofia S. Villar[1], David S. Robertson[1]

1 MRC Biostatistics Unit, University of Cambridge, Cambridge, United Kingdom, 2 Institute of Psychiatry, Psychology & Neuroscience, King's College London, London, United Kingdom

* xijin.chen@mrc-bsu.cam.ac.uk

## Abstract

When comparing the performance of multi-armed bandit algorithms, the potential impact of missing data is often overlooked. In practice, it also affects their implementation where the simplest approach to overcome this is to continue to sample according to the original bandit algorithm, ignoring missing outcomes. We investigate the impact on performance of this approach to deal with missing data for several bandit algorithms through an extensive simulation study assuming the rewards are missing at random. We focus on two-armed bandit algorithms with binary outcomes in the context of patient allocation for clinical trials with relatively small sample sizes. However, our results apply to other applications of bandit algorithms where missing data is expected to occur. We assess the resulting operating characteristics, including the expected reward. Different probabilities of missingness in both arms are considered. The key finding of our work is that when using the simplest strategy of ignoring missing data, the impact on the expected performance of multi-armed bandit strategies varies according to the way these strategies balance the exploration-exploitation trade-off. Algorithms that are geared towards exploration continue to assign samples to the arm with more missing responses (which being perceived as the arm with less observed information is deemed more appealing by the algorithm than it would otherwise be). In contrast, algorithms that are geared towards exploitation would rapidly assign a high value to samples from the arms with a current high mean irrespective of the level observations per arm. Furthermore, for algorithms focusing more on exploration, we illustrate that the problem of missing responses can be alleviated using a simple mean imputation approach.

## 1 Introduction

In healthcare, rapid progress in machine learning is enabling opportunities for improved clinical decision support [1–3]. In this context, Multi-Armed Bandit Problems (MABPs), which can be traced back to proposals for allocating patients in two-armed clinical trials [4], have seen a surge of interest in recent years [5, 6]. The goal in MABPs is to achieve a balance between allocating resources to the current best choice (exploitation) or to alternatives that

**Data Availability Statement:** This is a simulation study using synthetic data.: All relevant data are within the paper and code is available at https:// github.com/xijin-chen/plos-one.

**Funding:** This research was supported by the NIHR Cambridge Biomedical Research Centre (BRC1215-20014), the NIHR Maudsley Biomedical Research Centre at South London and Maudsley NHS Foundation Trust and King's College London. The views expressed in this publication are those of the authors and not necessarily those of the NHS, the National Institute for Health Research or the Department of Health and Social Care (DHCS). SSV received funding from the UK Medical Research Council (MC_UU_00002/15). DSR received funding from the Biometrika Trust and the UK Medical Research Council (MC_UU_00002/14). KML received funding from the National Institute for Health Research (NIHR Research Professorship, Professor Richard Emsley, NIHR300051). The funders had no role in study design, data collection and analysis, decision to publish, or preparation of the manuscript.

**Competing interests:** The authors have declared that no competing interests exist.

could potentially be a better choice (exploration). MABPs are a powerful framework for algorithms that make decisions over time under uncertainty, arising in a variety of application domains in recent decades, such as ad placement, optimizing seller's prices (also known as dynamic pricing or learn-and-earn), and internet routing and congestion control.

In the context of clinical trials, bandit algorithms fall within the larger class of response-adaptive designs, which allow the allocation of patients to depend on the previously observed responses in order to achieve experimental aims such as allocating more patients in the best arm. This paper will use the clinical trial setting as our main focus. Jacko [7] illustrates the formulated terminology of MABPs in various disciplines, and we follow the terminology that is widely used in the machine learning and clinical trial communities. For instance, 'algorithm' and 'arms' in MABPs correspond to 'design' and 'treatments' in biostatistics (see Table 1 in [7]).

A fixed (equal) randomization scheme that does not change with patient responses is still the most traditional and well-accepted allocation rule in clinical trial designs. However, there has been a surge of recent interest in response-adaptive designs [8]. The use of bandit algorithms is thus increasingly being considered in the context of adaptive clinical trials [9], potentially in combination with other adaptive features such as early stopping or sample size re-estimation [10]. There is a growing methodological literature discussing implementations of bandit models in various types or phases of clinical trial designs [5, 11–15]. However, it is difficult to point to examples where bandit algorithms have been used in clinical trials [5], except for algorithms based on a variant of the Thompson Sampling (TS) algorithm. Such examples in oncology include the well-known BATTLE trial [16] and the I-SPY 2 trial [17].

A key problem with the use of multi-arm bandits in practice is that the response information required to update the allocation of subsequent patients might not be available for many reasons. In clinical trials, enrolled patients may not complete the study and drop out without further measurements due to adverse reactions, death, or a variety of other reasons. In some healthcare settings, malfunctions in electronic health record systems commonly occur. It is common that the consequent *missing data* reduces the statistical power of a study and produces biased parameter estimates, leading to a reduction in efficiency and invalid conclusions [18].

The simplest analysis approach to missing data problems in clinical trials is to use only those cases that do not have any data missing in the variable(s) we are concerned with. This approach is referred to as 'complete case analysis' or 'listwise deletion' [19]. In response-adaptive designs, it is far more challenging to handle this problem than in traditional fixed designs as a result of the fully sequential nature and the cumulative impact of the allocation procedure [20]. In other words, it is not only an analysis problem at the end of a trial (as in a fixed design), but also a design problem in terms of the allocation procedure. The naive approach of ignoring missing data is thus suboptimal in terms of performance in the context of response-adaptive designs since the original intention of placing more patients in the best arm could be violated in this case.

To the best of our knowledge, this challenging problem has received limited attention in the context of response-adaptive designs [21], including bandit algorithms. One exception is Biswas and Rao [22], who implement regression imputation for a covariate-adjusted response adaptive procedure based on the available responses and associated covariates. A doctoral dissertation of Ma [20] concentrates on the doubly-adaptive biased coin designs in the presence of missing responses. A common approach in the literature is to assume that data are 'Missing at random' (MAR) [19], meaning that the probability of data being missing depends on the observed data but not the missing data. For responses that are MAR, a likelihood-based approach was proposed by Ma [20] to incorporate the information of covariates and prevent

inconsistent parameter estimates and undesirable allocation results. Sverdlov et al. [23] evaluate the impact of missing data on the proposed Longitudinal Covariate-Adjusted Response-Adaptive Randomization procedures with continuous responses, in terms of the targeted allocation proportions and the accuracy of final estimations of parameters of interest. Numerous simulation studies results of three different missing data analysis approaches show that the performance gets worse as the percentage of missing data increases. Finally, Williamson and Villar [24] propose using online imputation for the Forward-Looking Gittins Index rule [25] with normally distributed outcomes.

The issue of delayed outcomes in response-adaptive trials is closely related to the problem of missing responses. There have been a few proposals for dealing with delayed outcomes that simply treat those late-onset outcomes as missing data. A Bayesian data augmentation method has been proposed to impute those missing outcomes [26, 27]. Nevertheless, it is more common to distinguish between delayed responses and missing responses in most cases. Kim et al. [28] study the impact of delay in the outcome variables on the performance of doubly-adaptive biased coin designs with binary outcomes. It was shown that the corresponding performance could be improved by imputation based on a short-term predictor. Williamson et al. [29] identify the existing gap between the theory and clinical practice for Bayesian response-adaptive procedures by considering whether responses are intermediately available or delayed, respectively. However, these findings may not apply to the missing data problem in the context of response-adaptive designs, since missing values can be viewed as a very extreme form of delay where the outcomes would never be available [30]. In other words, the problem of missing data is distinct and has not received as much attention as the problem of delayed outcomes.

In the machine learning community, researchers have also realized that the key assumption of immediate reward after an action is taken may not hold in practice. For instance, Bouneffouf et al. [31] investigate contextual bandits with missing rewards, which reflect the outcome of the selected action in clinical trials, or whether an ad is clicked or not in recommender systems. They propose a modified version of the Upper Confidence Bound (UCB) algorithm, which imputes the missing rewards based on the available rewards from similar contexts. The setting of contextual bandits in the machine learning community corresponds to the covariate-adjusted response-adaptive designs in clinical practice, where the problem of missing data has been discussed the most together with associated covariates. For this reason, the proposed framework allows the use of available context information for future decision-making when some of the outcomes are missing. Missing data problem in the 'context' rather than in reward is more widely discussed, which is referred to as 'corrupted contextual bandits' [32]. Additionally, the problem with delays is also common in the field of machine learning, which might not be applicable in the extreme case of missing data problems as already mentioned above [33–38].

Here we revisit the problem of missing responses for the following reasons: (a) Most response-adaptive methods in clinical research are limited to response-adaptive randomization, which does not cover the more general case of *deterministic algorithms* favored in the machine learning community from the perspective of quality of care or patients benefit; (b) There is a gap in investing the impact of missing data for *finite sample sizes* since related research in machine learning applications focuses more on some asymptotic properties; (c) Related research is limited to *a limited range of response-adaptive designs or bandit algorithms*. Besides, the fact that MABP algorithms and response-adaptive designs are usually not compared makes it harder to make broad comparisons or useful recommendations. This indicates that further systematic investigation based on the fundamentals of bandit algorithms (i.e., the exploration-exploitation trade-off) is required; (d) Related work seldom accommodates different probabilities of missingness in different arms, indicating a more extensive

evaluation is in need; (e) Covariate-adjusted RAR and contextual bandits use side information, and it is unclear what the impact of missing data is on algorithms that do no use any covariates; (f) The closely related issue of delayed outcomes in trial designs has attracted a lot of attention and various solutions have followed. However, most of these proposed solutions do not necessarily apply to the problem of missing data. Our main contribution is to perform a comprehensive investigation concerning the fundamentals behind bandit algorithms, specifically, the exploration-exploitation trade-off, in the presence of missing responses. This goes beyond the few examples that have been looked at in isolation for specific response-adaptive designs or bandit models.

In Section 2, we first introduce the notations used in this paper, followed by a description of the allocation procedure in clinical trials in the presence of missing data. We then introduce some well-known bandit algorithms in the framework of MABPs. In Section 3, we consider different probabilities of missingness in the two arms, presenting a comprehensive simulation study of allocation results considering various scenarios either under the null or the alternative hypothesis. In addition, other patient-outcome metrics are also considered for illustration. In Section 4, we implement the mean imputation approach for missing binary responses, and discuss the impact of biased estimates in MABP on imputation. We conclude this paper with a discussion in Section 5.

## 2 General framework

We consider a general two-armed bandit model referring to the setting with binary responses in the presence of missing data. The two-armed bandit problem naturally arises as a sub-problem in some multi-armed generalizations and serves as a starting point for introducing additional problem features. Assuming responses are missing at random (MAR), the probability of being missing is the same within groups defined by the observed data (in our setting, the two treatment groups). Patients sequentially enrolled in a clinical trial will be assigned to the $k$ competing arms. In terms of the performance of different algorithms, we evaluate metrics of patient outcomes, including the proportion of patients assigned to the experimental arm ($p^*$) and the observed number of success (ONS), which we define below. Specifically, our main focus is on $p^*$ since the performance of different algorithms is more clearly seen. We evaluate the performance of different algorithms via simulation studies, which are performed using the R programming language [39].

### 2.1 Allocation procedure of bandit algorithms in the presence of missing data

The two-armed trial setting is a very common one in clinical trial designs. We denote the control arm by the subscript $k = 0$ and the experimental arm by $k = 1$, respectively. Assigned patients have missing responses with a probability $p_k^{\mathrm{m}} \leq 0.5$, given that it is rare that the majority of patients would be missing in the setting of healthcare. The total number of missing responses, successes and failures on arm $k$ at time $t$ are denoted by $M_{k,t}$, $S_{k,t}$ and $F_{k,t}$. The observed outcomes ($S_{k,t}$, $F_{k,t}$), with a true probability of success $p_k$, in turn have an impact on the next allocation due to the response-adaptive nature in bandit algorithms, while missing responses $M_{k,t}$ do not. The Bayesian feature is introduced to the algorithm by a uniform prior ($s_{k,0} = 1, f_{k,0} = 1$), which is the initial state that enables the first patients to be assigned when there are no observations. The uniform prior implies that an equal allocation (as in the fixed design) is initially used, but a different prior could also be used if appropriate.

The current state $x_{k,t} = (s_{k,0} + S_{k,t}, f_{k,0} + F_{k,t})$ after having observed $S_{k,t}$ and $F_{k,t}$ determines the subsequent patient allocation. Note that for each arm $k$, the number of assigned patients in

arm $k$ at time $t$ accounts for both the missing and observed data, namely, $N_{k,t} = S_{k,t} + F_{k,t} + M_{k,t}$. The fixed total trial size is expressed as $n$. The allocation procedure with bandit algorithms in the presence of missing data is attached in S1 Appendix. The operating characteristics we evaluate are defined as the proportion of patients assigned to the experimental arm ($p^* = \frac{N_{1,n}}{n}$) and the observed total number of success (ONS = $E[S_{0,n} + S_{1,n}]$). In the clinical trial context, we will typically be interested in testing a null hypothesis $H_0$: $p_0 = p_1$ against a one-sided alternative hypothesis $H_A$: $p_0 < p_1$.

## 2.2 Bandit algorithms

We classify several well-known bandit algorithms into three categories according to their exploration-exploitation trade-off. Details of these algorithms are summarized in Table 1, with further details given below.

- Randomized algorithms (TTS, RTS, and RPW): The patient at time $t$ is randomly assigned to arm $k$ with probability $\pi_{k,t}$.

- Deterministic algorithms (CB, GI, and UCB): The patient at time $t$ is deterministically assigned to the arm $k$ with the larger index values $I_{k,t}$, indicating the existence of a 'priority' value for sampling from one of the arms [40].

- Semi-randomized algorithms (RandUCB, RBI, and RGI): The patient at time $t$ is deterministically assigned to the arm with the larger index values $I_{k,t}$ as in deterministic algorithms. However, a stochastic element is involved in the computation of the index values $I_{k,t}$.

In the allocation procedure, the first assignment is based on the initial value ($\pi_{k,0}$ or $I_{k,0}$) and is an equal allocation for all algorithms, which is based on the choice of prior. This is achieved by allocation of the first patient via an allocation probability fixed at $\pi_{k,0} = 0.5$ in the case of randomized algorithms, or via equal index values $I_{k,0}$ in both arms in the case of deterministic or semi-randomized algorithms.

- *Tuned Thompson Sampling* (TTS): is not the original version of TS but a variant that has been more generally implemented [11]. TTS randomizes each patient to a treatment $k$ with a probability that is proportional to the posterior probability that treatment $k$ is the best arm given the accrued data. The best arm refers to the arm with the largest $p_k$, and it is assumed

**Table 1. Bandit algorithms defined by allocation probability $\pi_{k,t}$ or index value $I_{k,t}$.**

| Algorithm | $\pi_{k,t}$ or $I_{k,t}$ | Details | Initial value |
|---|---|---|---|
| TTS | $\pi_{k,t} = \frac{P(\max_t \ p_t = p_k)^c}{\sum_{k=1}^{K} P(\max_t \ p_t = p_k)^c}$ | $c = \frac{t}{2n}$ | $\pi_{k,0} = 0.5$ |
| RTS | $\pi_{k,t} = \frac{P(\max_t \ p_t = p_k)^c}{\sum_{k=1}^{K} P(\max_t \ p_t = p_k)^c}$ | $c = 1$ | $\pi_{k,0} = 0.5$ |
| RPW | $\pi_{k,t}$ is urn-based | initial urn with one ball for each arm $k$ | $\pi_{k,0} = 0.5$ |
| CB | $I_{k,t} = \hat{\mu}_{k,t}^{B}$ | $\hat{\mu}_{k,t}^{B} = \frac{s_{k,0} + S_{k,t}}{s_{k,0} + f_{k,0} + S_{k,t} + F_{k,t}}$ | $I_{k,0} = 0.5$ |
| GI | $I_{k,t} = \hat{\mu}_{k,t}^{G}$ | $\hat{\mu}_{k,t}^{G} = G[s_{k,0} + S_{k,t}][f_{k,0} + F_{k,t}]$ | $I_{k,0} = 0.8699$ |
| UCB | $I_{k,t} = \hat{\mu}_{k,t}^{B} + \beta \cdot \lambda_k(t)$ | $\beta = \sqrt{2\log(t)}, \ \lambda_k(t) = \frac{1}{\sqrt{s_{k,0} + f_{k,0} + S_{k,t} + F_{k,t}}}$ | $I_{k,0} = 0.5$ |
| RandUCB | $I_{k,t} = \hat{\mu}_{k,t}^{B} + Z_t \cdot \lambda_k(t)$ | $Z_t \sim f_X(x), \ \lambda_k(t) = \frac{1}{\sqrt{s_{k,0} + f_{k,0} + S_{k,t} + F_{k,t}}}$ | — |
| RBI | $I_{k,t} = \hat{\mu}_{k,t}^{B} + Z_t \cdot \lambda_k(t)$ | $Z_t \sim \text{Exp}(1/K), \ \lambda_k(t) = \frac{K}{s_{k,0} + f_{k,0} + S_{k,t} + F_{k,t}}$ | — |
| RGI | $I_{k,t} = \hat{\mu}_{k,t}^{G} + Z_t \cdot \lambda_k(t)$ | $Z_t \sim \text{Exp}(1/K), \ \lambda_k(t) = \frac{K}{s_{k,0} + f_{k,0} + S_{k,t} + F_{k,t}}$ | — |

that in the case of a tie ($p_0 = p_1$), the arm ($p_0$) would be considered as the best. The positive tuning parameter $c = \frac{t}{2n}$ is time-varying.

- *Raw Thompson Sampling* (RTS): is the 'raw' version of TS in its first proposal [4]. RTS is the most commonly used version in the machine learning community, which is different from the commonly used version, TTS, in the biostatistical community in terms of the tuning parameter [11]. The tuning parameter of TTS is fixed at $c = 1$ in RTS. When compared to the traditional fixed randomization (FR) with a fixed tuning parameter $c = 0$, RTS and TTS have different tuning parameters $c$.

- *Randomized Play-the-Winner* (RPW): is not a bandit algorithm but an urn-based model [41]. We include it since RPW has a long history in clinical trials methodology, and because its use helps explain why few response-adaptive clinical trials have occurred in practice. Indeed, the infamous Extracorporeal Circulation in Neonatal Respiratory Failure (ECMO) trial [42] used RPW to allocate patients. Due to the extreme treatment imbalance and highly controversial interpretation, the ECMO trial is regarded as a key example against the use of response-adaptive designs in clinical trials [43, 44]. The initial urn composition in this paper is the extreme case of one ball for each treatment. One would expect the allocation proportions to be less extreme when the initial urn composition is increased, as the urn will not favor the better treatment as highly.

- *Current Belief* (CB): allocates each patient to the treatment arm with the highest immediate estimate of reward $\hat{\mu}^B_{k,t}$, which is defined as the current highest mean posterior probability of success. With the only aim of exploitation, this myopic algorithm will tend to 'select' an arm before the trial is over. Once one arm is selected, all of the next enrolled patients will be assigned to this arm, and the index value of the unselected arm will not change anymore. One issue with CB is that the algorithm might make a wrong selection and make many allocations to the inferior arm. An early selection within the first few patients could be anticipated when the success probability $p_k$ is large.

- *Gittins Index* (GI): recovers the optimal solution to an infinite (discounted) MABP as obtained by Dynamic programming [5]. There is an upward adjustment in the index value $I_{k,t}$ in GI when compared to CB, referring to the uncertainty about the prospects of obtaining rewards from the arm [45]. This uncertainty corresponds to a decreasing exploration component, which decreases with more observations in the corresponding arm [46]. The value of the initial state $I_{k,0}$ of GI is larger than that of CB as a result of the upward adjustment. It is fixed at $I_{k,0} = 0.8699$ for GI in this paper, corresponding to a particular discount factor $d = 0.99$, the widely used value in the related bandit literature [5].

- *Randomized Belief Index* (RBI): is a semi-randomized approach with an index-based part $\hat{\mu}^B_{k,t}$ for exploitation and a random perturbation part $Z_t \cdot \lambda_k(t)$ for exploration. $Z_t$ is a random variable and $\lambda_k(t)$ is a decreasing sequence of positive constants. This means that most patients are assigned to the (current) superior arm, but some patients will still be assigned to the inferior arm in order to achieve exploration [47]. In other words, there is no aggressive determinism in RBI when compared to CB as a result of the additional perturbation part. The overall pattern of its index values $I_{k,t}$ is decreasing as a result of the decreasing exploitation component, accompanied by some fluctuations representing the random exploration part.

- *Randomized Gittins Index* (RGI): is a semi-randomized approach, which applies the random perturbation idea to the GI rule [47, 48]. Similar to RBI, the overall index values $I_{k,t}$ and the exploration component of the index values decrease over time.

- *Upper Confidence Bound Index* (UCB): is based on the principle of 'optimism in the face of uncertainty' [49]. It explores the arm with higher uncertainty because of higher information gain represented by the term $\beta \cdot \lambda_k(t)$, where $\beta = \sqrt{2\log(t)}$ and $\lambda_k(t) = \sqrt{\frac{1}{s_{k,0}+f_{k,0}+S_{k,t}+F_{k,t}}}$ reflects the confidence interval corresponding to the standard deviation in the estimation of reward $\hat{\mu}_{k,t}^B$. The more times a specific arm has been engaged before in the past, the greater the confidence boundary reduces towards the point estimate. If there are no observations in the $k$-th term, then $\hat{\mu}_{k,t}^B = 0$ and $\beta \cdot \lambda_k(t) \to \infty$, thus each arm is selected at least once. As evidence accumulates and the term $\beta \cdot \lambda_k(t)$ vanishes, the index values have an initial rapid increase for the first few patients but then decrease in both arms. There are many variants of UCB targeting various objectives in different disciplines [50].

- *Randomized Upper Confidence Bound* (RandUCB): is similar to RBI and RGI in terms of being a semi-randomized algorithm. RandUCB takes random perturbations [47, 48] for the computation of its index values. RandUCB is different from the UCB algorithm in terms of the exploration term, which is constructed by a random variable $Z_t$ sampling from a discrete distribution $f_X(x)$ [51]. In addition, exploration in the arm with higher uncertainty is realized by accounting for the confidence interval or standard deviation $\lambda_k(t)$ as in the UCB. The discrete distribution is supported on $M$ points over the interval $[L, U]$. Let $\alpha_1 = L \ldots, \alpha_M = U$ denote $M$ equally spaced points in $[L, U]$. We recover the UCB if $M = 1$ and $[L, U] = [\beta, \beta]$ hold. RandUCB with $M \to \infty$ approaches optimistic TS (one of the different versions of TS) with a Gaussian prior and posterior. In other words, both UCB and TS can be recovered as special cases of RandUCB. In this paper, we take $M = 20$ and $[L, U] = [0, 1]$ as an example for illustration. More details are discussed in Vaswani et al. [51]. Actually, the idea of RandUCB is the same as that used in RBI and RGI, which can be traced back to around 40 years ago [47].

In general, there is more than one strategy to solve the exploration-exploitation dilemma inherent in the MABP. For algorithms in this framework, the exploitation goal could be achieved with an immediate reward based on current observations. For the aim of exploration, one of the classic strategies is to account for a confidence level in the estimate, namely, **optimism in the face of uncertainty** [52]. The UCB algorithm is an example using this strategy that has attracted a lot of attention in the field of machine learning. Another common device in optimization problems is **randomized allocation**, which is widely used in the context of clinical trials. This includes the first well-known attempt at response-adaptive design, RTS. Besides, this stochastic setting has also motivated the 'Follow the Perturbed Leader' algorithms, which add random perturbations to the estimates of rewards of each arm prior to computing the current 'best arm' [53–55]. The semi-randomized algorithms, RBI and RGI [5], operate in the same way through randomly perturbing arms that would be selected by a greedy algorithm, which disregards any advantages of exploring. Thus, it allows mixtures of alternative decisions of the selected arms. RandUCB uses randomization to trade off exploration and exploitation, like RBI and RGI, and simultaneously it maintains optimism in the face of uncertainty in a similar way to UCB.

## 2.3 Sampling behaviour without missing data

Bandit algorithms work in different ways according to their exploration-exploitation trade-off. Fig 1 demonstrates the sampling behaviour using the results of a single simulation, for a specific scenario under the null. The trial size is fixed at $n = 200$, indicating a relatively small confirmatory clinical trial. In the case of a larger trial, the simulation results reflect what could

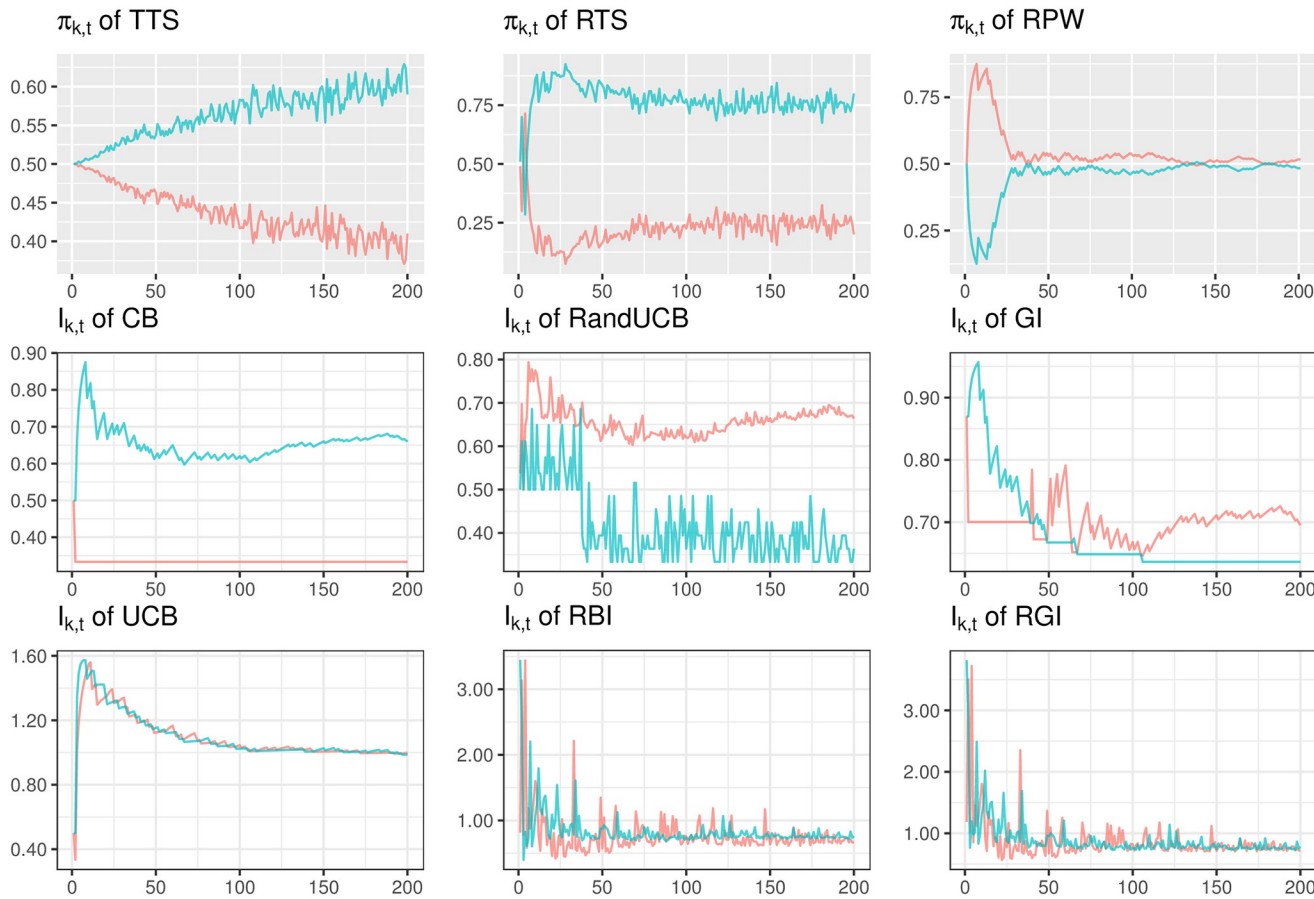

**Fig 1. Allocation procedure of bandit algorithms.** Performance of different bandit algorithms over a single simulation (with $p_0 = 0.7$, $p_1 = 0.7$, and $n = 200$) under the null. The two colors reflect different arms that could be regarded as the same under the null.

happen at its beginning. The true probabilities of success are fixed at $p_0 = p_1 = 0.7$, which might arise when testing two similarly efficacious vaccines in clinical trials. The assumption of identical arms also is relevant to some other machine learning applications, such as the problem of online allocation of homogeneous tasks to a pool of agents; a problem faced by many online platforms and matching markets. The regret performance of a bandit algorithm in terms of its arm-sampling characteristics are of great interest, not only in the 'large gap' (i.e., 'well-separated') instance but also the 'small gap' instance (i.e., 'worst-case'), where the gap corresponds to the small difference between the top two arm mean rewards [56]. The large treatment effects in the example in Fig 1 might not be common in general, but it can provide some intuitive thinking and theoretical perspective. We illustrate a counterpart for a scenario under the alternative in S5 Appendix because, in general, research in machine learning illustrates bandit performance under the alternative but the null case is rarely displayed.

In Fig 1, randomized algorithms (i.e., the first row) allocate patients with a probability $\pi_{k,t}$, while the others allocate patients to the arm with a larger $I_{k,t}$ value. Note that the scales on the y-axis of $I_{k,t}$ of deterministic and semi-randomized algorithms are different, and it is the relative difference between the $I_{k,t}$ in the two arms rather than the absolute value that determines allocation results.

The allocation probabilities $\pi_{k,t}$ of TTS show that it tends to 'select' an arm (i.e., allocate to that arm with a much higher probability) by the end of a large cohort of patients, even when

there is no treatment difference between these two arms. The random selection of one of these equal arms under the null has been discussed in the related discussion about bandit algorithms [57, 58]. A theoretical explanation of this imbalanced behaviour for Thompson Sampling under the null (i.e., despite the arms being statistically identical), also known as the 'incomplete learning' phenomenon, is provided in [56]. When compared with TTS, such a 'selection' under RTS happens earlier than that of TTS. RTS is also relatively more 'deterministic' than TTS due to the fixed tuning parameter, even though both are randomized algorithms. The urn-based algorithm RPW also illustrates some degree of 'determinism' in the first few patients. That is, one arm tends to be 'selected' at the beginning of the trial, as a result of the small initial urn size in our setting.

These randomized algorithms have a smaller chance of an early selection and never conduct a real selection in the finite sample size settings. In contrast, deterministic algorithms are more likely to truly select an arm, with patients always assigned to the selected arm from then on. An allocation skew could be anticipated if an early selection occurs. For CB and RandUCB, early selection is particularly noticeable, especially for CB. However, GI, although defined to be a deterministic rule, continues to explore in this instance where CB does not, as shown by the index values of both arms continuing to overlap throughout the trial. Meanwhile, RBI, RGI, and UCB are quite similar in a sense that the index values in both arms are close to to each other, although accompanied by fluctuations as a result of accounting for the uncertainty of the point estimate. This sampling behaviour is also referred to as 'complete learning', which was explained theoretically by [56], where UCB is taken as an example. In particular, UCB is able to discern statistical indistinguishability of the arm-means, and induce a 'balanced' allocation under the null when there is no treatment difference.

Since the results in Fig 1 are based on a single simulation under the null, a selection is neither incorrect nor correct as both arms are the same and there is no inferior or superior arm (all other costs being equal). A similar illustration of allocation probabilities $\pi_{k,t}$ or index values $I_{k,t}$ in a scenario under the alternative is given in S5 Appendix. The illustration in Fig 1 gives an insight into the potential consequences of the presence of missing data as shown in Section 3.1. An extensive simulation involving various scenarios follows in Section 3.3.

## 3 Impact of missing data

In this section, we investigate the impact of missing data on the expected proportion of patients assigned to the experimental arm $p^* = \frac{N_{1,n}}{n}$. On average, half of the patients ($E[p^*] = 0.5$) will be assigned to both arms under the null $H_0$, and more than half of the patients ($E[p^*] > 0.5$) will be assigned to the experimental arm under the alternative $H_A$. This aligns with the motivation for using response-adaptive designs in clinical practice. With a fixed trial size $n$, $p^*$ is determined by the number of patients assigned to the experimental arm $N_{1,n}$. Thus, the metric $p^*$ reflects how the allocation is affected in the presence of missing data.

In Section 3.1, we extend the discussion about Fig 1 taking missing data into account. That is, we explore how missing data in one arm affects $\pi_{k,t}$ or $I_{k,t}$ and hence the allocation results. In Section 3.2, we illustrate the impact of different probabilities of missingness in the two arms ($p_0^m$ and $p_1^m$) for an example under the null. In Section 3.2, we present a comprehensive simulation study under the null and the alternative.

### 3.1 Impact of missing data in one arm

For simplicity, suppose that missing data only occurs in arm $k$ and not arm $\tilde{k}$. If the outcome at time $t_1$ in arm $k$ is missing, the next allocation to arm $k$ at time $t_2$ cannot be updated and remains what it was for the last patient ($I_{k,t_2} = I_{k,t_1}$ or $\pi_{k,t_2} = \pi_{k,t_1}$). By contrast, since there is

no missing data at time $t_2$ in arm $\tilde{k}$, the value of $I_{\tilde{k},t_2}$ or $\pi_{\tilde{k},t_2}$ would be updated as it was supposed to do. Consequently, the subsequent allocations could be affected due to the missing data.

In addition, the impact of missing data on $\pi_{k,t}$ or $I_{k,t}$ is expected to differ across algorithms. For randomized and semi-randomized algorithms, the impact due to missing responses is relatively low as a result of exploration. This applies to both randomized algorithms (i.e., TTS and RTS) and semi-randomized algorithms (i.e., RBI and RGI). Similarly, algorithms taking uncertainty into account for allocations (i.e., UCB and GI) are also less affected by missing data due to their continued exploration.

By contrast, heavily exploitative algorithms (i.e., CB and RandUCB) are expected to be largely affected by missing data in terms of the selection. In the example in Fig 1 with large true probabilities of success ($p_0 = p_1 = 0.7$), the arm with missing outcomes is less likely to be selected. The reason is that the other arm without missing outcomes is more likely to have successful outcomes under such a large $p_k$, and $I_{k,t}$ of this arm will be updated to a larger value for the next allocation. As a consequence, these greedy or deterministic algorithms are less robust to the impact of missing data when compared with algorithms that are more explorative.

Besides, the changing patterns of $\pi_{k,t}$ and $I_{k,t}$ provide some additional insights on the impact of missing data. On the one hand, randomized algorithms (TTS and RTS) keep learning and $\pi_{k,t}$ of both arms change across the trial—starting from $\pi_{k,0} = 0.5$, $\pi_{k,t}$ values in these two arms become more distinguishable from each other with more enrolled patients and are always symmetric around 0.5. Consequently, the arm with larger $\pi_{k,t}$ values is more likely to be selected even if there is missing data in this arm. On the other hand, $I_{k,t}$ values generally decrease as the immediate reward $\hat{\mu}_{k,t}^B$ or $\mu_{k,t}^G$ decreases with more patients. Once the selection of an arm occurs, $I_{k,t}$ of the other arm will remain unchanged as the arm will not be selected anymore, see Fig 1. Consequently, the arm with missing data is more likely to be selected.

Considering both the impact of missing data in one arm and the changing patterns of $\pi_{k,t}$ or $I_{k,t}$ across a trial of different algorithms, it could infer the impact of missing data in one arm on allocation results, as shown in the simulation results in Section 3.2. Intuitively, for algorithms geared towards exploration, missing data in one arm slows down the progress of learning or exploration in randomized algorithms. More patients will thus be assigned to the arm with missing data to continue to explore. For algorithms geared towards exploitation, missing data in one arm impedes the decreasing trend in $I_{k,t}$ of the corresponding arm, which may thus have a larger index value than the other arm without missing data. Consequently, the arm with missing data is less likely to be selected, and fewer patients will be assigned to it.

## 3.2 An example of the impact of different types of missing data

In this section, we investigate the impact of having different probabilities of missingness in the two arms. Simulation results are based on $10^4$ independent replications of each algorithm, except for TTS, which has $10^3$ replications for computational reasons. We consider 36 combinations of missingness probabilities for the two arms, where $p_0^m$ and $p_1^m$ follow a sequence of discrete values ranging from 0 to 50% (0, 0.1, 0.2, 0.3, 0.4, 0.5). Fig 2 shows the simulation results for $E[p^*]$ of three representative algorithms from Table 1.

The value of $E[p^*]$ in each cell of Fig 2 corresponds to one of the combinations of missingness probabilities. The trial size is fixed at $n = 200$, and the true probabilities of success in this example ($p_0 = p_1 = 0.9$) are unrealistic in a healthcare setting, but are chosen to show the impact of missing data on allocations in an extreme case. A more comprehensive simulation study is given in Section 3.3, including scenarios under the alternative.

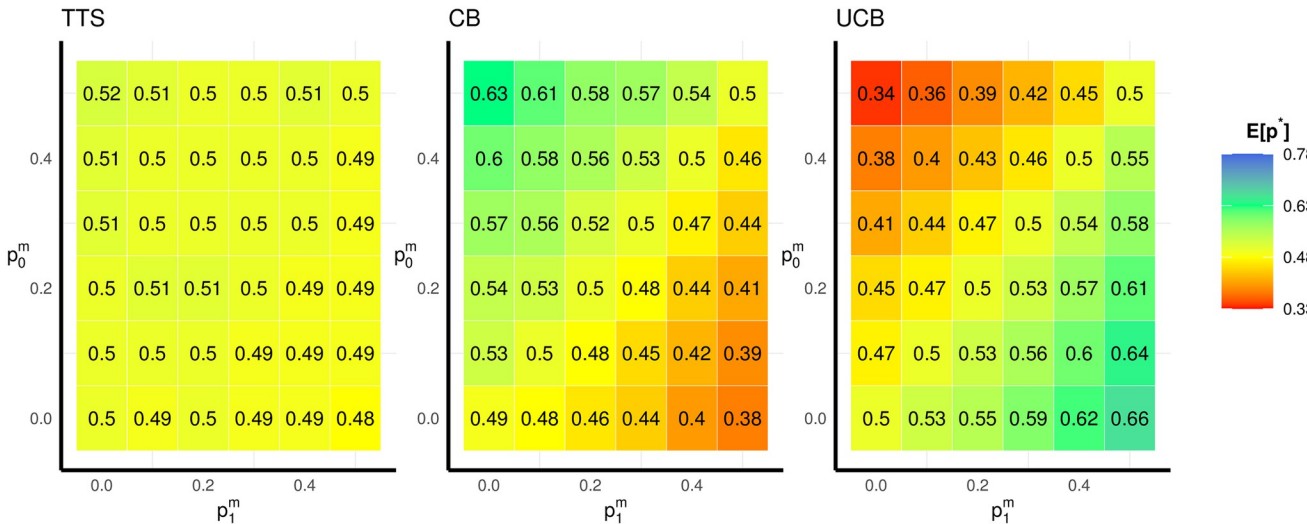

**Fig 2. Simulation results of $E[p^*]$ for TTS, CB and UCB under the null.** Expectations of $10^4$ replications are taken for CB and UCB and $10^3$ replications for TTS under different combinations of missingness probabilities, with $p_0 = p_1 = 0.9$ and $n = 200$.

As a comparison, fixed randomization always allocates approximately half of the patients to the experimental arm ($E[p^*] = 0.5$), regardless of the missingness probabilities. Similarly, the values of $E[p^*]$ for TTS under different probabilities of missingness are only slightly affected, as shown in Fig 2. However, this is not the case for CB and UCB, where allocations will be largely affected in the presence of missing data. Even though our assumption of MAR means that the missing data is non-informative, the corresponding impact on allocation results seems to be predictable if we know which arm has more available responses (equally fewer missing responses). To be specific, CB selects the experimental arm when this provides relatively more responses. Approximate 63% patients are assigned to the arm $k = 1$ when $(p_0^m, \ p_1^m) = (0, \ 0.5)$. In contrast, UCB assigns approximate 34% patients to the arm $k = 1$ in the same case with $(p_0^m, \ p_1^m) = (0, \ 0.5)$. One concern is that under the null, this can falsely promote the idea that the experimental arm (e.g., a new drug) is better because more patients are assigned to it and hence it will have relatively more observations (or vice-versa).

### 3.3 A comprehensive simulation study

**3.3.1 Simulation settings.** To investigate the impact of different probabilities of missingness, five scenarios under the null ($H_0$: $p_0 = p_1$) and seven scenarios under the alternative ($H_A$: $p_0 < p_1$) are investigated, as shown in Table 2. Scenarios under the null correspond to settings where the new treatment offers no benefit over control, which is important for clinicians and drug regulators to consider. In machine learning applications, the increase in performance goal fosters a focus on the alternative. In the context of clinical trials and other healthcare applications, simulation studies should consider both the null and alternative scenarios.

The twelve scenarios are taken from Rosenberger et al. [59], which uses the sample size that yields a power of 90% (under the alternative) when using equal allocation. Note that this consideration of power would not imply the same level of power for different response-adaptive rules. The setting of a fixed sample size may not be consistent with machine learning applications, which typically requires a large sample size. However, this setting is common for trial designs in clinical practice.

**Table 2. Simulation scenarios.**

| Scenario ($p_0$, $p_1$) | Trial size $n$ |
| --- | --- |
| S1 (0.10, 0.10) | 200 |
| S2 (0.30, 0.30) | 200 |
| S3 (0.50, 0.50) | 200 |
| S4 (0.70, 0.70) | 200 |
| S5 (0.90, 0.90) | 200 |
| S6 (0.10, 0.20) | 526 |
| S7 (0.10, 0.30) | 162 |
| S8 (0.10, 0.40) | 82 |
| S9 (0.40, 0.60) | 254 |
| S10 (0.60, 0.90) | 82 |
| S11 (0.70, 0.90) | 162 |
| S12 (0.80, 0.90) | 526 |

Simulation results are based on $10^4$ independent replications (except for TTS, which uses $10^3$ replications for computational reasons). We investigate 16 combinations of missingness probabilities, namely: (a) Equal probabilities of missingness in each arm ($p_0^m = p_1^m = p^m$), corresponding to the diagonal in Fig 2; (b) Missingness only occurring in the control arm ($p_0^m \in \{0, 0.1, 0.2, 0.3, 0.4, 0.5\}, \; p_1^m = 0$), corresponding to the leftmost column in Fig 2; (c) Missingness only occurring in the experimental arm ($p_1^m \in \{0, 0.1, 0.2, 0.3, 0.4, 0.5\}, \; p_0^m = 0$), corresponding to the bottom row in Fig 2. The problem of both differential and equal rates of missing data in two arms has attracted attention [60]. The impact of the other types of combinations of missing rates in two arms would correspond to interpolating between scenarios (a), (b), and (c).

**3.3.2 Under the null.** Fig 3 show the simulation results for $E[p^*]$ under the null. Each column corresponds to a bandit algorithm as described in Table 1. Simulation results of fixed randomization (FR) are also included as a reference. The five rows correspond to five scenarios under the null ($H_0$), as indicated in Table 2. The different line colors indicate results of $E[p^*]$ under different kinds of missing data: Grey lines correspond to the case of equal missingness probability in both arms; Blue lines correspond to missingness only in the control arm; Red lines correspond to missingness only in the experimental arm. Note that even though these scenarios are under the null, we distinguish the two arms ($k = 0$ and $k = 1$) with different colors.

When there is no missing data (i.e., when the probability of missingness is equal to zero), half of the patients are assigned to the experimental arm across all algorithms and all scenarios. With equal missingness, there is almost no impact on $E[p^*]$ under the null. In contrast, the impact due to missing data occurring in one arm ($p_0^m > 0$ or $p_1^m > 0$) can be large, as shown in Fig 3 and described in more detail below.

- *TTS* and *RTS*: these randomized algorithms are hardly affected by the impact of missing data occurring in one arm ($p_0^m > 0$, or $p_1^m > 0$), which is similar to the results of the standard FR. For RTS, the more exploitative nature introduced by the fixed tuning parameter $c = 1$ makes the allocation results slightly more affected by missing data under extreme scenarios (i.e., S1 (0.1, 0.1) and S5 (0.9, 0.9)) than TTS.

- *GI, UCB, RBI* and *RGI*: are explorative algorithms as a result of a randomized term or the uncertainty taken into account during allocations. As mentioned in Section 3.1, the index

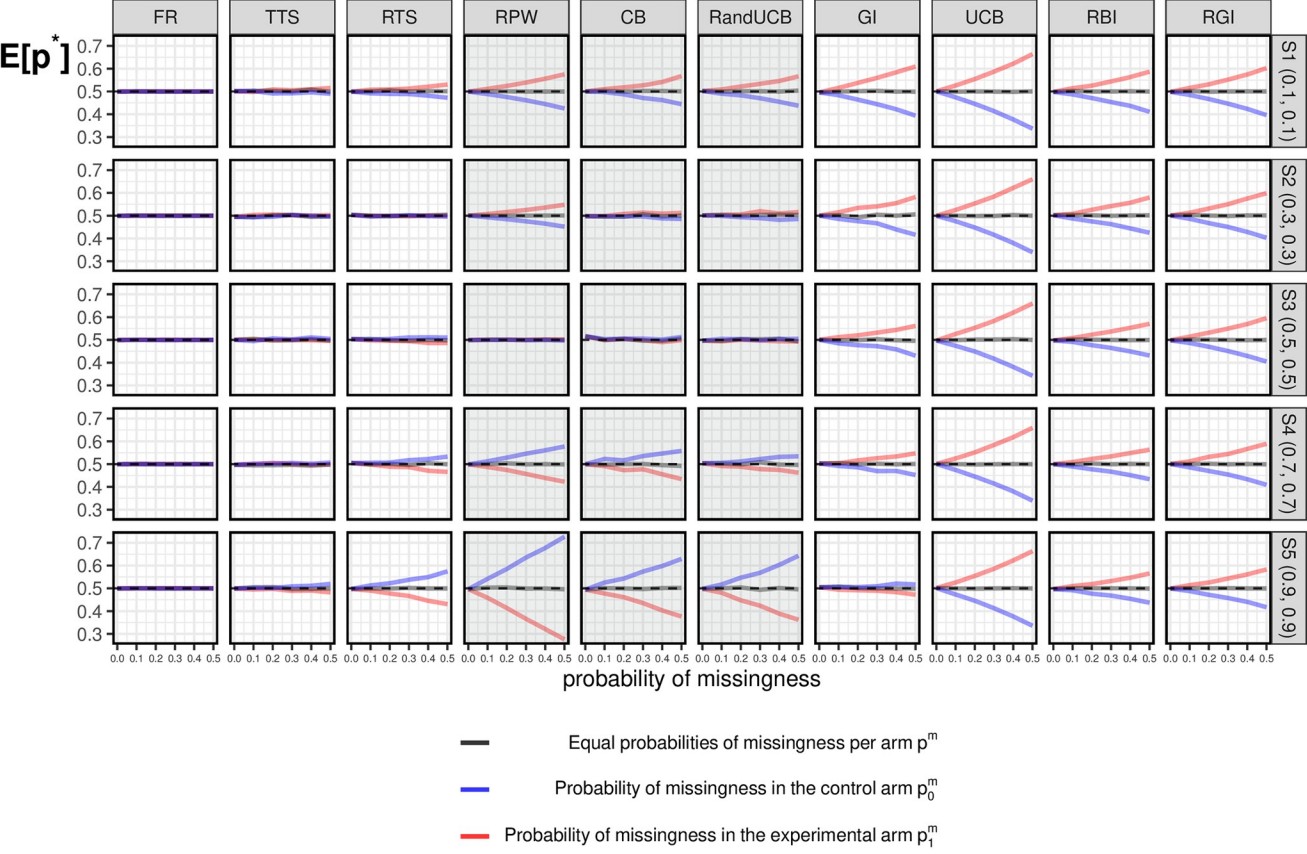

**Fig 3. Simulation results under the null.** Simulation results of $E[p^*]$ under the null for different missing data combinations. Grey lines correspond to the case of equal missingness probability in both arms; Blue lines correspond to missingness in the control arm; Red lines correspond to missingness in the experimental arm.

values $I_{k,t}$ may stop decreasing in the arm with missing responses. As a result, higher $I_{k,t}$ values in the arm with missing data can be expected, and more patients will be assigned to this arm for the purpose of exploration. This conclusion applies to all scenarios under the null. GI is an exception where the magnitude of the impact of missing data varies across different scenarios, since uncertainty towards the future varies under different $p_k$ values. Specifically, exploration dominates exploitation in most cases, while more exploitation and less exploration could be expected under larger $p_k$ since GI is a deterministic algorithm. The allocation results are thus hardly affected by missing data under S5 (0.9, 0.9), where there is a balance of exploration and exploitation with the existence of missing data.

By contrast, another deterministic algorithm that achieves exploration by taking uncertainty into account, UCB, performs differently from GI with the existence of missing data. Specifically, the goal of exploration is always dominant in UCB, and we can see consistent results under different scenarios. As a consequence, UCB assigns more patients to the arm with missing data. The allocation results can thus be strongly affected by missing data (steeper lines in Fig 3) when compared with semi-randomized algorithms (i.e., RBI and RGI) as well as randomized algorithms (i.e., TTS and RTS).

- *RPW*, *CB* and *RandUCB*: are algorithms focusing more on exploitation. This exploitative feature is not that obvious with a small value of $p_k$, as there is a relatively long period before

selection. By contrast, an early selection occurs with large values of $p_k$. In the case of small $p_k$, more allocations are required to the arm with missing values to decide on the selection. In the other case of large $p_k$ with early selections, if missing data occurs in an arm, the algorithm may opt to select the other arm instead. Consequently, the arm with missing data is less likely to be selected, and thus patients are less likely to be assigned to the corresponding arm. Note that under scenario S3 (0.5, 0.5), $E[p^*]$ is hardly affected by missingness. This is because of the balance between the impact of missing data before selecting the superior arm (exploration phase) and after selecting the superior arm (exploitation phase). For RandUCB, the values of $E[p^*]$ depend heavily on the parameter $M$, and as $M$ changes, the results would become similar to either TTS or UCB [51]. In the example displayed in Fig 3, the choice of $M = 20$ means that RandUCB is more exploitative, which is more clear in Fig 1.

### 3.3.3 Under the alternative.

Results for scenarios under the alternative are shown in Fig 4, where the experimental arm ($k = 1$) has larger true probabilities of success $p_k$ than the control arm ($k = 0$). More than half of the patients will be assigned to the superior arm when there is no missing data ($E[p^*] > 0.5$), except for the standard FR. This is one of the motivations for implementing response-adaptive designs in clinical trials. In most of the scenarios, RTS, GI, and RandUCB assign more than 80% of patients to the experimental arm. Similarly, TTS, UCB, RBI, and RGI are algorithms with relatively robust results across different scenarios. Approximately 70%-80% of patients are assigned to the experimental arm, regardless of how large $p_k$ are. The allocation results of CB depend heavily on the values of $p_k$. For example, more than 80% of patients are assigned to the experimental arm in S6 (0.1, 0.2), while approximately 60% of patients are assigned to the experimental arm in S12 (0.8, 0.9). This variation across different scenarios can also be seen for RPW, which allocates relatively more patients to the experimental arm in a scenario with small $p_k$ values.

In the presence of missing data, fewer patients will be assigned to the superior arm in general, when compared with when there is no missing data. Namely, we could observe a generally decreasing trend in all three lines corresponding to different kinds of missing data in most cases in Fig 4. This is because the presence of treatment differences always requires more patients to make the right decision regarding assignments, namely, to select the better arm. More details are given below.

- *TTS* and *RTS*: are slightly affected by missing data in a consistent way under the alternative scenarios. That is, fewer patients will be assigned to the superior arm no matter if the missing data occurs in both arms or only one of these arms. This feature allows the impact of missing data on TTS and RTS to be predictable in practice.

- *GI*, *UCB*, *RBI* and *RGI*: are largely affected by missing data in a consistent way under the alternative scenarios. Specifically, more missing data in the inferior arm ($p_0^m \geq 0$) results in more allocations to this arm and fewer allocations to the superior arm, as indicated by decreasing blue lines. For UCB, which is most affected by missing data, a high probability of missing data occurring in the control arm ($p_0^m = 0.5$) can even reduce the proportion of patients assigned to the experimental arm to approximately half of the patients under S6 (0.1, 0.2). In comparison, RBI and RGI are always less affected by missingness in the control arm. GI seems to be relatively robust in terms of the impact due to missing data, with more than 80% of patients assigned to the experimental arm in all scenarios.
  In terms of missingness in the experimental arm, the impact is less obvious than that in the control. It is even hard to observe the impact of the missingness in the experimental arm in most cases in Fig 4. The exception is that UCB allocates more patients to the experimental

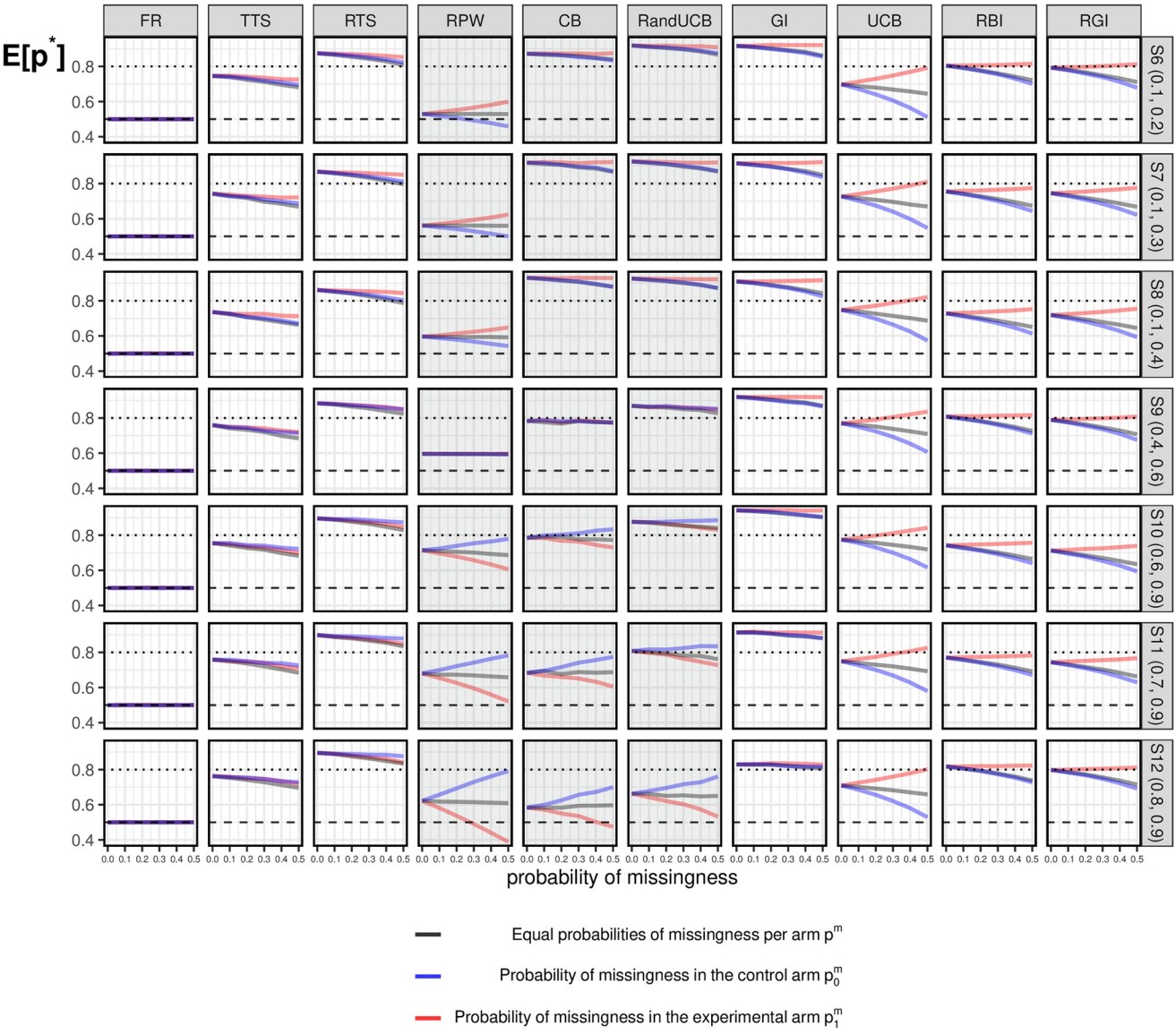

**Fig 4. Simulation results under the alternative.** Simulation results of $E[p^*]$ under the alternative for different missing data combinations. Grey lines correspond to the case of equal missingness probability in both arms; Blue lines correspond to missingness in the control arm; Red lines correspond to missingness in the experimental arm.

arm with a larger $p_1^m$. Lastly, the impact of equal missingness in both arms is in-between the impact of missingness in only one arm. This indicates that equal probabilities of missing data is only ignorable when there is no treatment difference.

- *CB*, *RPW* and *RandUCB*: perform differently across the scenarios since their deterministic nature depends on $p_k$. In scenarios with small $p_k$ (S6 (0.1. 0.2)—S8 (0.1, 0.4)), even though there is a slight reduction in $E[p^*]$ due to the impact of missing data in the control arm, more than 80% of patients are still assigned to the experimental arm. However, in scenarios with large $p_k$ (S10 (0.6, 0.9)—S12 (0.8, 0.9)), where an early selection occurs as a result of exploitation, these algorithms perform worse than in the scenarios with small $p_k$. In the extreme cases of CB and RPW, less than 50% of patients are assigned to the experimental

arm when there is a large probability of missing data in the experimental arm in S12 (0.8, 0.9). That is, the inferior arm might be incorrectly selected at the beginning of a trial, while the experimental arm never gets a chance to be explored after this incorrect selection.

In general, bandit algorithms are more likely to be affected by missing data than the standard FR. The existence of missing data can slow the progress of exploration in the corresponding arm. This means that more patients will ultimately be assigned to the arm with missing data, such that this arm could still be explored. In this case, a smaller impact could be expected with algorithms that have a greater focus on exploration. This corresponds to the fact that randomized algorithms (i.e., TTS and RTS) are less affected than semi-randomized algorithms (i.e., UCB, RBI, and RGI) and deterministic algorithms (i.e., CB, RandUCB, and GI). In contrast, for algorithms that are more exploitative rather than explorative, an early selection of an arm can occur under large $p_k$. Consequently, fewer patients will be assigned to the experimental arm if missing data occurs in that arm. This fact warrants careful attention when such algorithms are implemented under large $p_k$. $E[p^*]$ values of exploitative algorithms could be unexpectedly small because there are fewer observations in the experimental arm.

### 3.4 Observed number of successes

In clinical practice, another key metric of interest is the expected number of successes (ENS), namely, the oracle version based on the observed as well as the missing responses. Since in practice, ENS is the expected value of a quantity that will never be fully observed in the presence of missing data, the actual observed number of successes (ONS) might be more relevant to explore. S6 and S7 Appendices illustrate the simulation results for ONS under the twelve scenarios in Section 3.3. Since large numbers of missing values result in fewer successes, equal missingness always results in lower ONS values than missingness that occurred in one arm.

In scenarios under the alternative, the impact of missing data in the two arms is distinguishable from each other, and missing data in the experimental arm ($p_1^m > 0$) is a more serious problem. Across different algorithms in a given scenario, the impact of missingness in the experimental arm (the red lines in S7 Appendix) on the ONS varies across the algorithms. For the explorative algorithms, including randomized (i.e., TTS and RTS), semi-randomized (i.e., RBI and RGI), and deterministic algorithms (i.e., GI and UCB), more assignments to the arm with missing data are expected as discussed above. However, the ONS decreases dramatically with larger probabilities of missing data. In extreme cases with $p_1^m = 0.5$, we could even see smaller ONS in these algorithms than in FR. This indicates that assigning more patients to the experimental arm (due to missing data in this arm) cannot compensate for the missed observations. On the contrary, exploitative algorithms (i.e., RPW, CB, and RandUCB) will assign fewer patients to the arm with missing data, which is less likely to be incorrectly selected as the superior arm. It is interesting to see that the trend of decline of ONS with larger values of $p_1^m$ is far less dramatic than the explorative algorithms and even FR. This is reasonable considering that incorrect selection of the inferior arm as a result of the missing data could also lead to successful outcomes when $p_0$ is large. In the extreme case with $p_1^m = 0.5$, we could even see ONS in these algorithms larger than that in FR. This again encourages us to carefully consider the implementation of bandit algorithms in the presence of missing data.

## 4 Imputation results for the missing responses

To mitigate the impact due to missing data, we consider the use of mean imputation. Mean imputation is a kind of single imputation that replaces missing values with the mean value of that variable [61, 62]. Considering outcomes are missing at random, we intend to investigate

how a simple imputation technique would change the impact of missingness in terms of the allocation results of bandit algorithms. For each patient at time $t$, we compute an estimate $\hat{p}_{k,t} = \frac{S_{k,t}}{S_{k,t}+F_{k,t}}$ of the true probability of success $p_k$. A random value is then drawn from a Bernoulli distribution with estimated probability of success $\hat{p}_{k,t}$ to replace the missing outcome. In the case of $S_{k,t}+F_{k,t} = 0$, the corresponding $\hat{p}_{k,t}$ is not available. We instead use a default initial value $\hat{p}_{k,0} = 0.5$, indicating that the treatment has an a-priori success probability of 50% in this case. The allocation procedure with bandit algorithms after the missing data is imputed is given in S2 Appendix.

## 4.1 Under the null

Fig 5 shows the simulation results of $E[p^*]$ for scenarios under the null both with and without mean imputation for missing data. We use solid lines to represent the results without mean imputation, while dashed lines represent the results with mean imputation. For equal probabilities of missing data per arm, the results of $E[p^*]$ are hardly affected by missing data. That is, approximately half of the patients are assigned to the experimental arm on average, and the mean imputation approach does not make any differences. In the cases of missingness in only

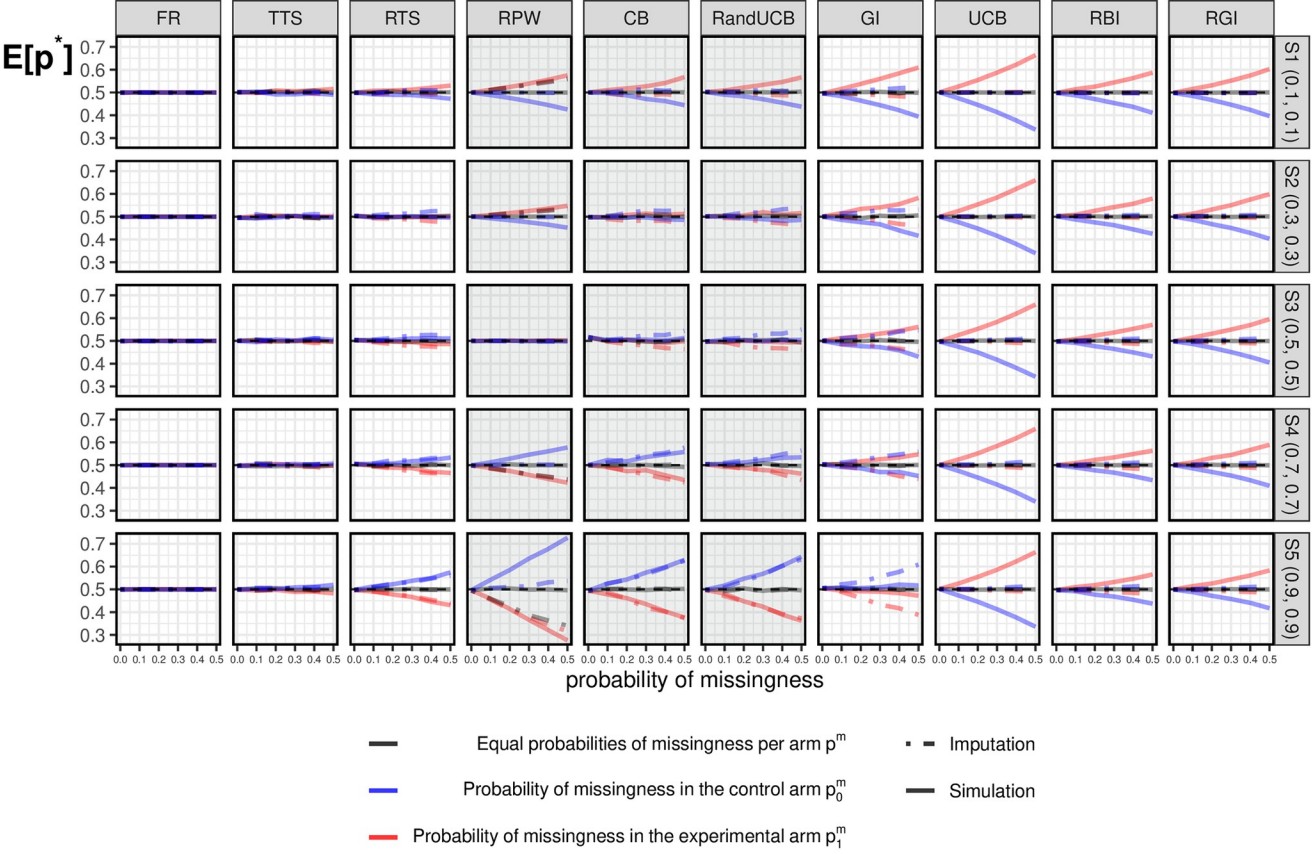

**Fig 5. Imputation results under the null.** Imputation results of $E[p^*]$ under the null for different missing data combinations with initial value $\hat{p}_{k,0} = 0.5$. Grey lines correspond to the case of equal missingness probability in both arms; Blue lines correspond to missingness in the control arm; Red lines correspond to missingness in the experimental arm. Solid lines correspond to the results without mean imputation, while the dashed lines correspond to the results with mean imputation.

one arm, imputation works differently in terms of the inherent exploitation-exploration trade-off and specific scenarios, as discussed below.

- *Exploration*: more explorative algorithms, including randomized algorithms (i.e., TTS and RTS), semi-randomized algorithms (i.e., RBI and RGI), and deterministic algorithms (i.e., UCB), assign more patients to the arm with missing data. Mean imputation could largely solve this problem, and this applies across all scenarios (S1 (0.1, 0.1)—S5 (0.9, 0.9)), with the exception of RTS in S5 (0.9, 0.9), where the exploitative nature dominates the explorative nature as explained in Section 3.3.

- *Exploitation*: In scenarios with small $p_k$ (i.e., S1 (0.1, 0.1) and S2 (0.3, 0.3)), there is no early selection. In this case, we observe a similar but limited impact of missingness as in the case of explorative algorithms. That is, there are more allocations in the arm with missing data. Consequently, mean imputation could help with allocation skew in the way it works for the explorative algorithms. However, this is not the case under large $p_k$ (i.e., S4 (0.7, 0.7) and S5 (0.9, 0.9)), where an early selection occurs and the algorithms exhibit great determinism in the allocation procedure. In this case, the impact due to missing data cannot be mitigated by mean imputation, and sometimes it even exaggerates this impact. For instance, GI assigns fewer patients to the experimental arm with imputation for $p_1^{\mathrm{m}}$ in S5 (0.9, 0.9), which is even worse than without imputation.

## 4.2 Under the alternative

Fig 6 illustrates simulation results of $E[p^*]$ for scenarios under the alternative both with and without mean imputation for missing data. In general, since more patients are required to make the right allocation decision (i.e., select the superior arm) under the alternative, relatively fewer patients are assigned to the experimental arm in the presence of missing data when compared with results in a same scenario without missing data (as discussed in Section 3.3.3).

- *Exploration*: the impact of missing data could largely be mitigated for randomized (i.e., TTS and RTS), semi-randomized (i.e., RBI and RGI) and deterministic algorithms (i.e., UCB) as for scenarios under the null in Fig 5. One exception is RTS in scenario S12 (0.8, 0.9), where it exhibits a more exploitative nature due to the fixed tuning parameter.

- *Exploitation*: mean imputation does not work well to mitigate the impact due to missing data, especially when there is an early selection with larger $p_k$. In this case, missing data occurring in the experimental arm is a serious problem with a substantial reduction of assigned patients to this arm. GI is a special case that is less likely to be affected by missing data across all of these scenarios. A particular concern with GI is that mean imputation in a scenario with large $p_k$ (S10 (0.6, 0.9)—S12 (0.8, 0.9)) could even greatly exaggerate the impact of missing data. In S12 (0.8, 0.9), with the existence of a large proportion of missing data ($p_m = 0.5$ or $p_1^m = 0.5$), approximately 90% of patients will be assigned to the experimental arm, while this drops to only 80% after imputation.

## 4.3 The impact of biased estimates on mean imputation

The implemented mean imputation approach is based on the currently estimated probabilities of success $\hat{p}_{k,t}$, which is known to be biased in the context of adaptive sampling [5, 63–65]. Villar et al. [5] empirically demonstrate the presence of negative bias of the sample means for a number of multi-armed bandit algorithms in a finite sample setting via a simulation study. Nie et al. [64] formally provide two conditions called 'Exploit' and 'Independence of Irrelevant

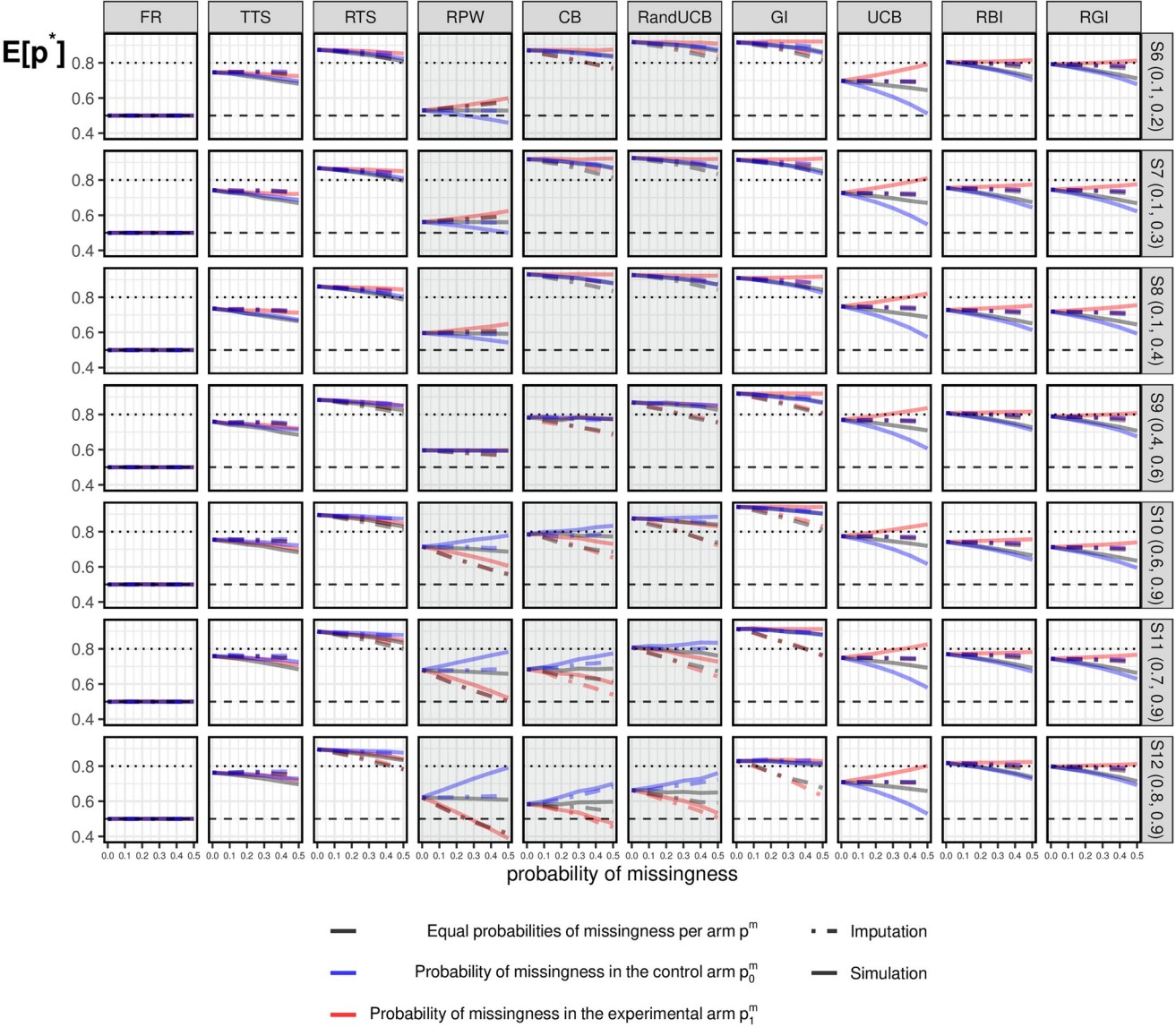

**Fig 6. Imputation results under the alternative.** Imputation results of $E[p^*]$ under the alternative for different missing data combinations initial value $\hat{p}_{k,0} = 0.5$. Grey lines correspond to the case of equal missingness probability in both arms; Blue lines correspond to missingness in the control arm; Red lines correspond to missingness in the experimental arm. Solid lines correspond to the results without mean imputation, while the dashed lines correspond to the results with mean imputation.

Options' (IIO) under which the negative bias of sampling mean exist. Shin et al. [65] provides a more general and comprehensive characterization of the sign of the bias of the sample mean in multi-armed bandits, which discuss the same context as ours (adaptive sampling in finite sample) and some other cases (adaptive stopping and adaptive selection).

To interpret the results of using mean imputation of missing data in this setting, the magnitude of the bias matters more than the sign of the bias. Bowden and Trippa [63] derived an exact formula to quantify the bias for randomised data dependent rules in a finite sample as shown in Eq (1). This simple characterisation indicates that the bias increases in magnitude as the dependency of $\hat{p}_{k,t}$ and $N_{k,t}$ increases. For this reason, FR guarantying a zero covariance

between these two is expected to have zero bias of sample mean estimate. In contrast, in the common case where optimistic adaptive randomization is used to direct more allocations towards arms that appear to work well, $N_{k,t}$ will be variable and positively correlated with $\hat{p}_{k,t}$. Notice that the derivation of the bias in [63] applies to randomized algorithms, while [65] then extends this formula to more general cases of MABP (i.e. non-randomized algorithms).

$$\frac{Cov[N_{k,t},\ \hat{p}_{k,t}]}{E[N_{k,t}]} = p_k - E[\hat{p}_{k,t}] = -\text{Bias}(\hat{p}_{k,t}) \tag{1}$$

Consequently, we could expect a larger negative bias under the exploitation feature where there is a large covariance between $\hat{p}_{k,t}$ and $N_{k,t}$ because of the rather myopically 'optimistic' sampling behaviour. In contrast, algorithms with more exploration features will see relatively smaller bias and be less reactive to current mean estimates. This aligns with the simulation results in Fig 4 in [5]. Similar simulation results for bias of the investigated algorithms in this paper are attached in S12 Appendix for scenarios under the null and in S13 Appendix for scenarios under the alternative. It is substantially more complicated to discuss the negative bias of treatment effect estimates accounting for the three different missing data combinations. However, the first intuitive conclusion we can make is that larger bias is expected in algorithms with a great amount of short term exploitation. For instance, CB and RandUCB show large negative bias in all scenarios. RTS and GI seem to be more biased when the treatment effect is larger (e.g., S4 (0.7, 0.7)—S5 (0.9, 0.9) and S9 (0.4, 0.6)—S12 (0.8, 0.9)). The magnitude of the negative bias is relatively small in TS, UCB, RBI, and RGI, which are relatively more explorative within the size of the experiment. Further investigation of how the bias varies between the control and treatment arms (under the alternative), as shown in S13 Appendix, is another interesting direction for research for in the future.

Consequently, for more explorative algorithms, a single success or failure does not have a large impact on the value of $I_{k,(t+1)}$ or $\pi_{k,(t+1)}$ of the subsequent allocation. Besides, the negative bias of $\hat{p}_{k,t}$ is relatively small for these algorithms. Thus, mean imputation based on $\hat{p}_{k,t}$ could work to mitigate the impact due to missing data. In contrast, for the exploitative algorithms, an early selection could be substantially affected by a single success or failure. In addition, the negative bias of $\hat{p}_{k,t}$ is relatively large in these cases. As a consequence, imputation based on an inaccurate estimate $\hat{p}_{k,t}$ may fail to mitigate the impact of missing data on exploitation. In other words, mean imputation with missing data replaced by a single value does not work for more short term exploitative algorithms that are highly sensitive to the success or failure of each allocation.

We also tried conducting imputation in two settings: 1. with a large initial value $\hat{p}_{k,0} = 0.9$ (see S8 and S9 Appendices); and 2. only after the first observation (see S10 and S11 Appendices). In either of these cases, the problem of biased estimates lead to undesirable characteristics of $E[p^*]$.

Since the simple mean imputation based on biased estimates works differently under exploitation and exploration features, some adjustments of the balance of the exploration-exploitation trade-off might help. For instance, imputation for missing data might work for RandUCB with a different turning parameter (e.g., using a larger value of $M$), which could introduce more 'randomization'. Besides, the idea of an additional perturbed component to the current estimate of the reward provides a similar way of thinking about exploration. In this sense, a randomized version of deterministic algorithms (i.e., RBI to CB, RGI to GI and RandUCB to UCB) should be amenable to an imputation approach. Apart from this, taking uncertainty into account in the computation of $I_{k,t}$ also supports exploring both arms, and the idea of deterministic algorithms GI and UCB aligns with this when compared with the myopic and

greedy algorithm CB. In these cases, a simple mean imputation could work for the impact of missing data. This idea of more exploration in deterministic algorithms has gained popularity in related domains [53, 54, 66].

Given the problem of negative bias of the treatment effect estimates, there have been a number of proposals in the literature showing to obtain unbiased or bias-reduced estimates. Nie et al. [64] explore data splitting and a conditional MLE as two approaches to reduce this bias, using considerable information about (and control over) the data generating process. Meanwhile, Deshpande et al. [67] use adaptive weights to form the 'W-decorrelated' estimator, a debiased version of OLS. They attempt to use the data at hand, along with coarse aggregate information on the exploration inherent in the data generating process. In addition, Dimako-poulou et al. [68] propose the Doubly-Adaptive Thompson Sampling, which uses the strengths of adaptive inference to ensure sufficient exploration in the initial stages of learning and the effective exploration-exploitation balance provided by the TS mechanism. In this paper, we did not try these novel debiasing approaches to obtain unbiased estimates for imputation. However, these proposals provide natural future directions to improve the performance of imputation in the presence of missing data.

In conclusion, mean imputation could reduce the problem of oversampling the arm with more missing data, which is a result of the explorative nature of some algorithms. However, it does not work for the problem of undersampling the arm with more missing data in the case of short term exploitative algorithms. The reason is that for these algorithms that perform differently in various scenarios, the impact of missing data varies according to this sampling behaviour. Since information about the true scenario is not accessible in reality, it is hard to use a simple mean or some other single imputation approach to mitigate the impact of missing data.

## 5 Conclusion

The problem of missing data in the implementation of machine learning algorithms in contexts such as healthcare applications is commonly overlooked. We demonstrate that bandit algorithms are not robust in the presence of missing data in terms of expected performance, e.g. in terms of expected patient outcomes. The omission of this practical issue, as well as the naive approach of ignoring it in related works, may prevent the realization of the potential advantages of widespread use of bandit algorithms in healthcare settings, including clinical trial designs. Through a simulation study, we have investigated the impact of missing data on randomized, semi-randomized, and deterministic bandit algorithms for scenarios under the null and the alternative. In terms of machine learning implementations, scenarios under the null are often ignored. However, these results under the null allow us to focus on the impact of missing data and isolate the impact of treatment differences on allocation results. In addition, it suggests that different scenarios need to be taken into account when choosing an algorithm in healthcare implementations. Specifically, for scenarios under the alternative, the potential consequences could be serious as the presence of missing data may reverse our preferences for the experimental or superior arm and opt for the control or inferior arm in extreme cases.

In this paper we focus on a practical setting of an experiment with a finite number of patients. The setting of a finite number of actions and outcomes has also gained attention in some other machine learning applications [69]. The investigation of different combinations of missing probabilities in two arms (i.e., equal missingness or missingness occurring in only one arm) should be given more attention. For scenarios under the null, the impact of equal missingness probabilities in both arms can be simply ignored. The practical implication is that if there is non-informative missing data (i.e., MAR) and even if the true success rates $p_k$ are

unknown, we can infer what the impact of missingness on different bandit algorithms will be. This applies to scenarios under the null and the alternative in general. Most importantly, there is a distinction between the results for explorative and exploitative algorithms, which corresponds to the inherent exploration-exploitation trade-off in a finite sample. Since deterministic algorithms are sometimes preferred over randomized algorithms from the perspective of quality of care [6, 70, 71], our findings show the need for careful thinking about the potential impact of missing data. More generally, incorporating randomization, perturbation, or accounting for uncertainty in deterministic algorithms for the aim of exploration can bring some advantages, see further discussions of optimism in the field of online learning [53, 54, 66].

The general rule guiding the impact of missing data on bandit algorithms, considering all the factors mentioned above, is in the way to balance exploration and exploitation. Explorative algorithms encourage more assignments in the arm with missing data (being optimistic in the face of uncertainty as less sample arms will have larger variance), while exploitative algorithms are less likely to select the arm with missing data (being more myopic and reacting more to the current mean than to the current variance). In the latter case of exploitative algorithms, this impact consequently depends on whether an early selection occurs and hence the success rates $p_k$ in both arms. The impact of missing data on the use of machine learning in healthcare has not received much attention so far, especially in settings other than clinical trial designs. An important message of this paper is a call for actively considering the impact of missing data when selecting a bandit algorithm for a specific application in which missing data is likely to occur.

Instead of simply ignoring missing data, imputation replaces missing values in the response-adaptive procedure. Assuming outcomes are MAR, we have investigated conducting mean imputation based on the current estimated probability of success $\hat{p}_{k,t}$. Our results show that the mean imputation approach supports the exploration procedure in heavily explorative algorithms, so that the impact of missing data on explorative algorithms could be mitigated by imputation. Specifically, there would be no additional assignments in the arm with missing data if imputation is implemented. This applies to the cases of explorative algorithms, including randomized algorithms (i.e., TTS and RTS), semi-randomized algorithms (i.e., RBI and RGI), and deterministic algorithms (i.e., UCB). By contrast, in the case of more exploitative algorithms, the impact of missing data is different for scenarios with small or large success rates ($p_k$). As a consequence, the problem of missing data could not be alleviated by a mean imputation approach based on current estimates $\hat{p}_{k,t}$. Worse still, in some extreme cases, the impact of missing data could even be exaggerated. This is especially obvious with large treatment effects $p_k$. The negative bias that has been well recognized and proved for MABPs also helps to explain the imputation results. That is, exploitative algorithms are associated with a large covariance between the number of allocations in one specific arm and the corresponding estimated probability of success, and thus exhibit a large negative bias of less sampled arms. As a result, mean imputation can be based on biased estimates which does not alleviate the issues caused by the presence of missing data.

We note that the conclusions about imputation results are limited to the assumption of MAR, which is untestable. In cases where the MAR assumption does not hold, which is a common setting in machine learning applications, algorithms may not be able to avoid selection bias. This is of particular importance in many areas of machine learning applications. Hence, the assumption underlies our conclusions where fairness is a concern, which is an emerging area in machine learning of how to ensure that the outputs of a model do not depend on sensitive attributes in a way that is considered unfair [72, 73]. This leads to an important caveat of

improper representation and exclusion of fairness issues in the implementation of machine learning algorithms. As future research, we could extend the current single imputation approach to more advanced methods of imputation [31, 74] or more complex missing data settings, such as when patient covariates are taken into account. In addition, rather than only considering a single approach to the missing data problem, appropriate forms of sensitivity analysis are important and necessary to assess the validity of these assumptions and the robustness of the results [23].

Finally, discussions about delayed outcomes have received increasing attention both in the community of machine learning [33, 75] and biostatistics [38, 76, 77] in recent years. In this paper, we assumed that all of the missing outcomes would never become available. This missing data problem could be viewed as an extreme form of delay, and an interesting avenue for future work would be to account for both missing data and delayed outcomes together.

## Supporting information

**S1 Appendix. The allocation procedure with bandit algorithms in the presence of missing data.**
(PDF)

**S2 Appendix. The allocation procedure with bandit algorithms and imputation for missing data.**
(PDF)

**S3 Appendix. Simulation results of $E[p^*]$ for TTS, CB and UCB under the null.** Expectations of $10^4$ replications are taken for CB and UCB and $10^3$ replications for TTS under different combinations of missingness probabilities, with $p_0 = p_1 = 0.7$ and $n = 200$.
(TIF)

**S4 Appendix. Simulation results of $E[p^*]$ for TTS, CB and UCB under the alternative.** Expectations of $10^4$ replications are taken for CB and UCB and $10^3$ replications for TTS under different combinations of missingness probabilities, with $p_0 = 0.7$, $p_1 = 0.9$, and $n = 200$.
(TIF)

**S5 Appendix. Performance of different bandit algorithms over a single simulation under the alternative ($p_0 = 0.7$, $p_1 = 0.9$, and $n = 200$).** Blue line represent $\pi_{k,t}$ or $I_{k,t}$ in the experimental arm and red lines represent that in the control arm.
(TIF)

**S6 Appendix. Observed number of success in the presence of missing data under the null.** The impact of missing data is similar under different algorithms under the null: the impact due to the equal probabilities of missing data per arm is larger than the impact of only having missingness in the control or experimental arm.
(TIF)

**S7 Appendix. Observed number of success in the presence of missing data under the alternative.** The impact of missing data on ONS varies among different algorithms. Generally, bandit algorithms outperform FR when there is no missing data. Equal missingness is a more serious problem than missingness only occurring in the experimental arm, which in turn is a much more serious problem than missingness only occurring in the control arm.
(TIF)

**S8 Appendix. Imputation results under the null with initial value $\hat{p}_{k,0} = 0.9$.** Imputation results of $E[p^*]$ under the null for different missing data combinations with initial value

$\hat{p}_{k,0} = 0.9$. Grey lines correspond to the case of equal missingness probability in both arms; Blue lines correspond to missingness in the control arm; Red lines correspond to missingness in the experimental arm. Solid lines correspond to the results without mean imputation, while the dashed lines correspond to the results with mean imputation.
(TIF)

**S9 Appendix. Imputation results under the alternative with initial value $\hat{p}_{k,0} = 0.9$.** Imputation results of $E[p^*]$ under the alternative for different missing data combinations with initial value $\hat{p}_{k,0} = 0.9$. Grey lines correspond to the case of equal missingness probability in both arms; Blue lines correspond to missingness in the control arm; Red lines correspond to missingness in the experimental arm. Solid lines correspond to the results without mean imputation, while the dashed lines correspond to the results with mean imputation.
(TIF)

**S10 Appendix. Imputation results under the null, with imputation starting after the first observation.** Imputation results of $E[p^*]$ under the null for different missing data combinations, with imputation starting after the first observation. Grey lines correspond to the case of equal missingness probability in both arms; Blue lines correspond to missingness in the control arm; Red lines correspond to missingness in the experimental arm. Solid lines correspond to the results without mean imputation, while the dashed lines correspond to the results with mean imputation.
(TIF)

**S11 Appendix. Imputation results under the alternative, with imputation starting after the first observation.** Imputation results of $E[p^*]$ under the alternative for different missing data combinations, with imputation starting after the first observation. Grey lines correspond to the case of equal missingness probability in both arms; Blue lines correspond to missingness in the control arm; Red lines correspond to missingness in the experimental arm. Solid lines correspond to the results without mean imputation, while the dashed lines correspond to the results with mean imputation.
(TIF)

**S12 Appendix. Bias of treatment effect after $n$ = 200 patients.** Simulation results of $\hat{p}_{k,t} - p_{k,t}$ under the null for different missing data combinations. We illustrate the result for one of the two equal arms. Grey lines correspond to the case of equal missingness probability in both arms; Blue lines correspond to missingness in the control arm; Red lines correspond to missingness in the experimental arm.
(TIF)

**S13 Appendix. Bias of treatment effect after $n$ = 200 patients.** Simulation results of $\hat{p}_{k,t} - p_{k,t}$ under the alternative for different missing data combinations. We illustrate the result for one of the two equal arms. Grey lines correspond to the case of equal missingness probability in both arms; Blue lines correspond to missingness in the control arm; Red lines correspond to missingness in the experimental arm. Solid lines correspond to the bias in the treatment arm, while the dashed lines correspond to the bias in the control arm.
(TIF)

## Author Contributions

**Formal analysis:** Xijin Chen, Kim May Lee, Sofia S. Villar, David S. Robertson.

**Investigation:** Kim May Lee, Sofia S. Villar, David S. Robertson.

**Methodology:** Kim May Lee, Sofia S. Villar, David S. Robertson.

**Project administration:** Sofia S. Villar.

**Resources:** Kim May Lee, Sofia S. Villar, David S. Robertson.

**Software:** Xijin Chen.

**Supervision:** Kim May Lee, Sofia S. Villar, David S. Robertson.

**Validation:** Kim May Lee, Sofia S. Villar, David S. Robertson.

**Visualization:** Xijin Chen.

**Writing – original draft:** Xijin Chen.

**Writing – review & editing:** Kim May Lee, Sofia S. Villar, David S. Robertson.

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
