## [Decision Letter · Decision Letter 0]

12 May 2022

PONE-D-22-10081Some performance considerations when using multi-armed bandit algorithms in the presence of missing dataPLOS ONE

Dear Dr. Chen,

Thank you for submitting your manuscript to PLOS ONE. After careful consideration, we feel that it has merit but does not fully meet PLOS ONE’s publication criteria as it currently stands. Therefore, we invite you to submit a revised version of the manuscript that addresses the points raised during the review process.

I received one review from an expert in the field.  The reviewer has a generally positive impression of your manuscript, but lists several points that need to be addressed.  if you feel you can address these concerns, then I invite you to submit a revision.

We look forward to receiving your revised manuscript.

Kind regards,

Darrell A. Worthy, Ph.D

Academic Editor

PLOS ONE

Journal Requirements:

3.Thank you for stating the following financial disclosure: 

"This research was supported by the NIHR Cambridge Biomedical Research Centre (BRC1215-20014), the NIHR Maudsley Biomedical Research Centre at South London and Maudsley NHS Foundation Trust and King’s College London. The views expressed in this publication are those of the authors and not necessarily those of the NHS, the National Institute for Health Research or the Department of Health and Social Care (DHCS). SSV received funding from the UK Medical Research Council (MC_UU_00002/15). DSR received funding from the Biometrika Trust and the UK Medical Research Council (MC_UU_00002/14). KML received funding from the National Institute for Health Research (NIHR Research Professorship, Professor Richard Emsley, NIHR300051)."

Reviewers' comments:

Reviewer's Responses to Questions

**Comments to the Author**

1. Is the manuscript technically sound, and do the data support the conclusions?

Reviewer #1: Yes

2. Has the statistical analysis been performed appropriately and rigorously? 

Reviewer #1: N/A

3. Have the authors made all data underlying the findings in their manuscript fully available?

Reviewer #1: Yes

4. Is the manuscript presented in an intelligible fashion and written in standard English?

Reviewer #1: Yes

5. Review Comments to the Author

Reviewer #1: High-level overview: This paper evaluates empirically standard multi-armed bandit algorithms in view of practical considerations involving missing data in the context of A/B testing for clinical trials. The "Missing at Random" (MAR) model is posited for missing data. The authors focus on the standard two-armed model (treatment/control) under Bernoulli outcomes and the "fraction of total assignments to treatment" and the "total number of successes" as the performance metrics of interest. Under the MAR model, some binary outcomes for one or both arms may be (independently) unobservable potentially with different probabilities, but the total number of samples from each is known at all times; this is the "missing data" problem investigated in the paper. Sensitivity/robustness of different algorithms to MAR data (in terms of stated performance metrics) is assessed empirically in a hypothesis testing setup under the null of "zero treatment effect." The same exercise is also repeated for the "mean imputation" model for unobservable (missing) outcomes.

Scope: This is an empirical paper that focuses on the measurement/benchmarking of the impact of MAR data on key performance metrics of commonly used bandit algorithms for A/B testing.

Review of extant literature: I think the following articles (and appropriate references therein) should be cited since they are related to the theme of this paper:

1. Why adaptively collected data have negative bias and how to correct for it [Nie et al., AISTATS 2018].

2. Are sample means in multi-armed bandits positively or negatively biased? [Shin et al., NeurIPS 2019].

3. Accurate inference for adaptive linear models [Deshpande et al., ICML 2018].

4. Online Multi-Armed Bandits with Adaptive Inference [Dimakopoulou et al., NeurIPS 2021].

Aforementioned works investigate theoretically the directions, causes, implications and mitigations for biases in bandit algorithms resulting from sample-adaptivity, and are highly relevant to this paper, especially in the context of mean imputation for missing data. Cited references may be able to provide a theoretical explanation for many of the empirical observations made in this submission. In addition, reference [56] (bibliography) provides a theoretical explanation for the "imbalanced" behavior of RTS (Raw Thompson Sampling) under the null, observable also in experimental results in this submission. The same reference also provides an explanation for the behavior of UCB1 (Auer et al., 2002) under the null; these aspects should be elucidated in detail in view of the numerical experiments conducted in this submission.

Miscellaneous: Line 243 -- Shouldn't it be "t" instead of "T" in the expression for \\beta? Otherwise the algorithm would correspond to a different version of UCB.

6. PLOS authors have the option to publish the peer review history of their article (what does this mean?). If published, this will include your full peer review and any attached files.

Reviewer #1: No

---

## [Author Response · Author response to Decision Letter 0]

15 Jul 2022

We thank the reviewer for pointing out the metioned theoretical references. We agree that including these are a very helpful addition and highly relevant to our paper, in particular in explaining the results seen when using mean imputation for missing data. We have now incorporated all of the reference into our paper in a new section 4.3 (‘The impact of biased estimates on mean imputation’), as summarised in the letter of 'Response to Reviewers'. This section is the major revision we have incorporated and is a direct response to the reviewer’s insightful feedback that allowed us to better explain our results. Note also we have updated our abstract in an attempt to better present our main observations in light of our reading of the references suggested by the reviewer.

1. Nie et al. (2018) prove that the bias of the sample mean for any fixed arm and at any fixed time is negative when the sampling strategy satisfies two conditions called ‘Exploit’ and ‘Independence of Irrelevant Options’ (IIO). Besides, they suggest two ways targeting the biased estimate via modifying the data collection procedure. We discuss the mean imputation results with the explanations of negative bias in our revised manuscript and we mention this paper in Line 674, 676, 743.

2. Shin et al. (2019) theoretically discuss that in many typical MAB settings, we should expect sample means to have two contradictory sources of bias: negative bias from ‘optimistic sampling’ and positive bias from ‘optimistic stopping/choosing’. This not only provides broader discussion

than the contexts in Nie et al. (2017) and our submission, which could be some directions for future research, but also this work extends the formula for negative bias given in Bowden and Trippa (2017) that only applied to randomised data adaptive sampling rules. The formula in both references give us some insights on the magnitude of bias in different multi-armed algorithms. We discuss this in combination with our experimental results to provide additional and new interpretations in Line 674, 678, 691 of the revised manuscript.

3. Deshpande et al. (2018) discuss the simple case of multi-armed bandits without covariates, where the ordinary least squares estimates correspond to computing sample means for each arm. They propose to decorrelate the OLS estimator. Even though this is not implemented to do imputation in our submission, we discuss how this could be an avenue for future investigation

for the imputation of the missing data in Line 745 of the revised manuscript.

4. Dimakopoulou et al. (2021) proposed the Doubly-Adaptive Thompson Sampling (DATS) by harnessing the strengths of adaptive inference estimators to ensure sufficient exploration in the initial stages of learning and the effective exploration-exploitation balance provided by the TS mechanism. This debiasing technique is not used in our submission, but we discuss how this could be a way of handling the problem of biased estimate for some specific algorithms (i.e., TS) in Line 748 of the revised manuscript.

We also thank the reviewer for highlighting Kalvit and Zeevi (2021) (Reference [56]), which discusses the sampling behaviour of TS and UCB under the ‘large gap’ (i.e., ‘well-separated’) and ‘small gap’ (i.e., ‘worst-case’) instances. The latter setting matches the scenarios under the null in our experimental investigations. For this reason, this reference helps to explain the ‘imcomplete sampling’ (i.e., ‘random selection’ in our submission) of TS from a theoretical

perspective. This behaviour is different from the ‘complete learning’ behaviour of UCB (i.e. inducing a ‘balanced’ allocation under the null), which has also been seen in our experimental results. We have modified the related discussions to include this reference as an explanation of the sampling behaviour of TS and UCB in Line 300, 318, 335 of the revised manuscript.

---

## [Decision Letter · Decision Letter 1]

25 Aug 2022

Some performance considerations when using multi-armed bandit algorithms in the presence of missing data

PONE-D-22-10081R1

Dear Dr. Chen,

We’re pleased to inform you that your manuscript has been judged scientifically suitable for publication and will be formally accepted for publication once it meets all outstanding technical requirements.

I sent your manuscript back to the original reviewer, and they felt all their comments had been adequately addressed.  Therefore, I am happy to accept your manuscript for publication.

Kind regards,

Darrell A. Worthy, Ph.D

Academic Editor

PLOS ONE

Additional Editor Comments (optional):

Reviewers' comments:

Reviewer's Responses to Questions

**Comments to the Author**

1. If the authors have adequately addressed your comments raised in a previous round of review and you feel that this manuscript is now acceptable for publication, you may indicate that here to bypass the “Comments to the Author” section, enter your conflict of interest statement in the “Confidential to Editor” section, and submit your "Accept" recommendation.

Reviewer #1: All comments have been addressed

2. Is the manuscript technically sound, and do the data support the conclusions?

Reviewer #1: Yes

3. Has the statistical analysis been performed appropriately and rigorously? 

Reviewer #1: N/A

4. Have the authors made all data underlying the findings in their manuscript fully available?

Reviewer #1: Yes

5. Is the manuscript presented in an intelligible fashion and written in standard English?

Reviewer #1: Yes

6. Review Comments to the Author

Reviewer #1: (No Response)

7. PLOS authors have the option to publish the peer review history of their article (what does this mean?). If published, this will include your full peer review and any attached files.

Reviewer #1: No

---

## [Editor Report · Acceptance letter]

2 Sep 2022

PONE-D-22-10081R1 

Some performance considerations when using multi-armed bandit algorithms in the presence of missing data 

Dear Dr. Chen:

I'm pleased to inform you that your manuscript has been deemed suitable for publication in PLOS ONE. Congratulations! Your manuscript is now with our production department. 

Kind regards, 

on behalf of

Dr. Darrell A. Worthy 

Academic Editor

PLOS ONE